# Non-Additive Time-Series Forecasting via Cross-Decomposition and Linear Attention

## Abstract

Many multivariate forecasters model additive effects well but miss non-additive interactions among temporal bases, variables, and exogenous drivers, which harms long-horizon accuracy and attribution. We present TIM, an all-MLP forecaster designed from the ANOVA/Hoeffding target: the regression function is decomposed into main effects and an orthogonal interaction component. TIM assigns the interaction to a DCN-style cross stack that explicitly synthesizes bounded-degree polynomial crosses with controllable CP rank, while lightweight branches capture main effects. Axis-wise linear self-attention (time and variables) transports information without increasing polynomial degree and maintains linear time and memory complexity. A decomposition regularizer encourages orthogonality and yields per-component attributions. We establish degree and rank guarantees and a risk identity showing that the additive error gap equals the energy of the interaction subspace. On long-term multivariate benchmarks, TIM matches or surpasses state-of-the-art accuracy with transparent cross-term explanations.

## 1 Introduction

Long-sequence time series forecasting (LSTF) is central to applications such as weather prediction (Ahamed & Cheng, 2024), traffic management (Zhao, 2019), equipment monitoring (Zhou et al., 2021), and power consumption analysis (Hebrail & Berard, 2006). In these domains, models must capture both short-term fluctuations and long-range dependencies under non-stationary dynamics. Transformers, with their attention mechanisms, have recently driven progress by capturing complex temporal and cross-variable patterns. However, their quadratic time and memory complexity hinders scalability, and their performance often degrades on very long horizons, limiting robustness in precisely the settings where LSTF is most needed.

Motivated by these limitations, the community has explored a broader landscape of architectures. RNNs (Damaševičius et al., 2024; De et al., 2024) provide sequential efficiency but struggle with long-term memory. State-space models (SSMs) (Rangapuram et al., 2018; Auger-Méthé et al., 2021; Newman et al., 2023; Orvieto et al., 2023) revive latent recurrence with improved memory, but at the cost of intricate parameterization and domain-specific priors. Surprisingly, compact MLPs (Yi et al., 2024; Zhang et al., 2022; Yeh et al., 2024; Zeng et al., 2023) often achieve competitive results, though they usually lack explicit mechanisms for modeling multivariate interactions. These approaches, despite their diversity, reveal a persistent trade-off between scalability, expressivity, and architectural simplicity.

Alongside architectural families, several strategies seek to enhance forecasting. Decomposition-based methods (Wu et al., 2021; Zhou et al., 2022; Bandara et al., 2020; Hao & Liu, 2024) aim to disentangle trend and seasonal components, while channel-independent versus channel-dependent (CI/CD) designs (Liang et al., 2023; Nie et al., 2023; 2024) seek a balance between parameter efficiency and cross-channel expressivity. Many recent designs can in fact be viewed as implicitly emphasizing additive effects (e.g., per-variable trends or seasonalities), while leaving interaction effects (cross-variable and cross-temporal dependencies) less explicitly modeled. Yet these strategies provide only partial remedies and remain vulnerable to non-stationarity, evolving seasonalities, or sensor noise. These limitations highlight the need for a principled decomposition that explicitly separates main and interaction effects within a lightweight and robust forecasting framework.

We present *Time-series Interaction Machine* (**TIM**), an all-MLP forecaster explicitly aligned with the ANOVA/Hoeffding decomposition. Unlike prior models that handle additive and interaction effects implicitly, TIM separates them by design: lightweight branches capture main effects, while a DCN-style cross stack models the orthogonal non-additive interaction subspace. Concretely:

- **Efficiency.** TIM achieves linear time and memory complexity via axis-wise linear self-attention (time and variables) combined with DCN-based feature crossing, while keeping parameters comparable to compact MLP baselines.
- **Expressivity.** The cross stack synthesizes bounded-degree polynomial crosses among temporal bases, lagged variables, and exogenous inputs (Theorem 2). Group-wise masking ensures higher-order terms correspond to genuine cross-channel interactions (Theorem 3), and low-rank cross maps provide controllable CP rank (Corollary 1).
- **Interpretability & Stability.** A decomposition regularizer enforces orthogonality between main effects and interactions, enabling component-wise attribution and stabilizing training. Our risk identity further shows that the error gap of additive models equals the energy of the interaction subspace (Appendix Theorem 1).

In summary, TIM is designed to be *efficient*, *expressive*, and *interpretable*. By aligning its architecture with a principled decomposition of additive and interaction effects, it scales linearly, explicitly models both main trends and cross-variable dependencies, and grounds its design in theory, offering a robust and transparent solution for long-sequence forecasting.

## 2 PRELIMINARIES AND METHODS

### 2.1 MATHEMATICAL LOGIC AND DECOMPOSITION ALIGNMENT

We design the architecture to match the population ANOVA/Hoeffding decomposition in Equation equation 2. The main-effect modules parameterize $f_t^B, f_t^X, f_t^Z$, and the cross branch parameterizes the *non-additive* interaction components $f_t^{BX}, f_t^{BZ}, f_t^{XZ}, f_t^{BXZ}$. The cross stack yields bounded-degree polynomials (Theorem 2); with group-wise masks it generates interaction-only monomials (Theorem 3). Its degree-$d$ coefficients admit a CP/Khatri–Rao factorization with controllable rank (Corollary 1). These properties support order-wise and path-wise attribution and justify a decomposition regularizer that encourages orthogonality between temporal bases and interaction components.

### 2.2 AXIS-WISE LINEAR SELF-ATTENTION: FUNCTIONAL RELATION AND ROLE

We employ linear self-attention along the time axis and the variable axis to propagate information efficiently. Let queries/keys/values be $q_t, k_s, v_s \in \mathbb{R}^{d_h}$ per head and adopt a positive feature map $\phi : \mathbb{R}^{d_h} \to \mathbb{R}^r$. Linear attention computes

$$\text{Attn}(q_t, \{k_s, v_s\}_s) = \frac{\phi(q_t)^\top \left( \sum_s \phi(k_s) v_s^\top \right)}{\phi(q_t)^\top \left( \sum_s \phi(k_s) \right)}.$$

This mechanism efficiently propagates information along both temporal and variable axes. Importantly, as we show in Appendix B.3, axis-wise linear self-attention preserves the polynomial degree and mask constraints imposed by the cross stack, thereby acting as a stable transport operator without introducing new higher-order interactions. Having established this theoretical safeguard, we now view linear attention from a functional perspective: the context equals a kernel smoother with kernel $K(t, s) = \phi(q_t)^\top \phi(k_s)$ and normalization.

**Along time**, this induces a *data-adaptive, positive semidefinite* temporal kernel that mixes the window $\{t - L+1, \ldots, t\}$ in linear time via associative pre-aggregation $\sum_s \phi(k_s)$ and $\sum_s \phi(k_s)v_s^\top$.

**Along variables**, an analogous kernel mixes channels at each time. This factorization preserves linear time–memory complexity and complements the cross branch: (i) it transports low-/mid-order cross features across distant lags and channels without increasing polynomial degree; (ii) it improves gradient flow and information routing across long contexts; and (iii) it leaves the *order/rank* control to the cross stack, keeping identifiability intact.

## 2.3 GENERAL ARCHITECTURE

According to Li et al. (2023), our model, like many others, consists of three key components: RevIN, a reversible normalization layer; an MLP; and a linear projection layer that generates the final prediction results. In our proposed architecture, MLP is used to extract time series features. In subsequent modules, we will employ a decomposition method to enable our model to learn from multivariate interaction features, temporal characteristics of the time series, and decomposed components, respectively. The full architecture of TIM can be found in Figure 1.

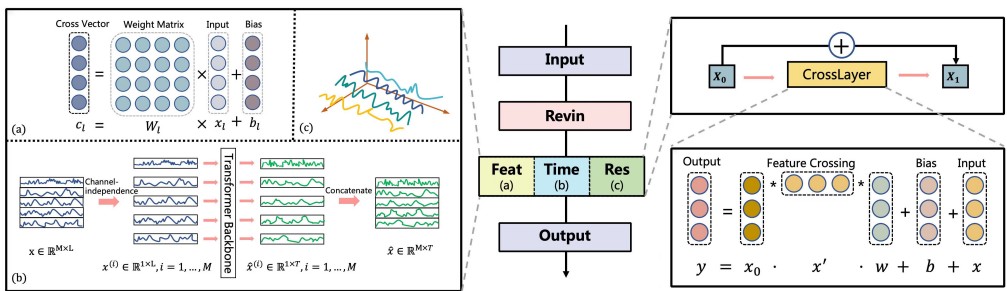

Figure 1: Overall TIM Architecture. TIM consists of three key components: Feat Fusion, which extracts multivariate interaction features; Time Fusion, which models temporal shifts across time points; and a residual modelling component for temporal, multivariable, or noise effects. The outputs of these modules—$X_{Feat}$, $X_{Time}$, and $X_{Res}$—are combined to produce the final forecast, which is then passed through a linear projection layer and inverse-transformed via RevIN to scale it back to the target domain for the prediction horizon.

## 2.4 FUSION ARCHITECTURE IN TIM

Most LSTF systems rely on seasonal–trend *data-side* decompositions Bandara et al. (2020); Hao & Liu (2024); Wu et al. (2021); Zeng et al. (2023), which can be brittle on long, nonstationary sequences. We instead adopt a *model-side* decomposition aligned with the ANOVA/Hoeffding target: (i) main temporal effects, (ii) non-additive multivariate interactions, and (iii) a residual adapter. TIM realizes these with three fusion branches that share the same backbone but differ in input–output shapes.

- **Feat Fusion (interaction branch).** A DCN-style cross stack generates bounded-degree crosses among lagged variables, temporal bases, and exogenous inputs, yielding multivariate interaction features $X_{\text{feat}}$. Group-wise masks enforce that higher-order terms represent genuine cross-group interactions, capturing the *non-additive (interaction) component*.

- **Time Fusion (additive branch).** Per-time snapshots are processed along the time axis (linear self-attention or MLP fusion) to capture temporal shifts and *main additive effects*, yielding $X_{\text{time}}$.

- **Res Fusion (residual branch).** The remaining variation is modeled by a lightweight adapter that captures residual structure not explained by main or interaction effects, producing $X_{\text{res}}$. A decomposition regularizer encourages orthogonality among the three branches.

The model composes these components as

$$Y = X_{\text{feat}} + X_{\text{time}} + X_{\text{res}},$$

followed by a linear projection to the prediction horizon and an inverse RevIN transform back to the target domain. This design keeps interactions explicit and attributable, captures temporal evolution without fragile pre-decomposition, and maintains linear time–memory complexity through axis-wise processing.

## 2.5 Linear Attention Gated Unit for Feature Extraction

We provide a detailed analysis of the TIME FUSION, FEAT FUSION, and RES FUSION modules for time-series feature extraction. Although these modules differ in input–output organization, they all share the same feature extraction algorithm based on the *Linear Attention Gated Unit (LAGU)*.

Specifically, the TIME FUSION and RES FUSION modules operate on per-variable temporal sequences with tensors $X \in \mathbb{R}^{T \times H}$, where $T$ is the sequence length and $H$ the hidden dimension. By contrast, the FEAT FUSION module processes feature-wise snapshots with tensors $X \in \mathbb{R}^{F \times H}$, where $F$ denotes the number of variables. Despite these architectural differences, all three branches apply the same gated linear attention mechanism for representation learning.

---

**Algorithm 1** Fusion Architecture for Time Fusion, Feat Fusion and Res Fusion

---

**Require:** Input $X_0 \in \mathbb{R}^{F \times H}$. Number of Layers $N$. Sigmoid function denoted as $\sigma$. Concatenate function denoted as cat. Linear layer mappings from the dimension 2*dim to dim, denoted as combine and gate.
**Ensure:** Output $X_L \in \mathbb{R}^{F \times H}$
1: Initialize $X_i = X_0$
2: **for** $i = 1$ to $N$ **do**
3:     Compute $y = W_i X_i + b_1$ {$y \in \mathbb{R}^{F \times H}$}
4:     Compute residual res $= \tilde{W}_i(X_i - y) + b_2$ {res $\in \mathbb{R}^{F \times H}$}
5:     Concatenate $x_{\text{cat}} = \text{cat}(y, \text{res})$ {$x_{\text{cat}} \in \mathbb{R}^{F \times 2H}$}
6:     Apply ELU activation $x_{\text{elu}} = \text{ELU}(x_{\text{cat}}) + 1$
7:     Gate and combine $x_{\text{out}} = \text{combine}(x_{\text{elu}}) \cdot \sigma(\text{gate}(x_{\text{elu}}))$
8:     Update $X_1 = X_0 \cdot x_{\text{out}}$ {$x_1 \in \mathbb{R}^{F \times H}$}
9:     Apply Dropout $X_1 = \text{Dropout}(X_1)$
10:    Update $X_i = X_i + X_1$ {$x_i \in \mathbb{R}^{F \times H}$}
11: **end for**
12: **return** $X_i$

---

Time Fusion, Feat Fusion and Res Fusion share the same architecture described in Algorithm 1. The design objective of the TIME FUSION module is to leverage linear attention for extracting temporal features, as supported by our mathematical derivations. In conventional linear attention, weights are derived by computing similarities between Query and Key vectors. However, they are obtained through element-wise multiplication with the input $X_0$. This formulation enables linear self-attention while progressively mapping temporal sequences from the latent space of the source domain into the state space of the target domain. Without residual connections, the update rule can be expressed as:

$$X_N = X_0 + \sum_{i=1}^{N} D(X_0 \circ (\text{ELU}(W_i X_{i-1} + b_i) + 1)), \tag{1}$$

where $\circ$ denotes the Hadamard product (element-wise multiplication) and $D(\cdot)$ represents dropout.

Dropout can be interpreted as an implicit gating mechanism that randomly discards neurons, akin to the suppression of irrelevant information in attention. Although it does not explicitly implement gating, its effect resembles that of attention weight distributions. To make the model more responsive to state changes, we introduce a residual structure that facilitates the capture of temporal transitions.

As shown in Algorithm 1, the input–output dimensions of the TIME FUSION and RES FUSION modules remain consistent, with tensors $X \in \mathbb{R}^{T \times H}$, where $T$ is the sequence length and $H$ the hidden dimension. In contrast, the FEAT FUSION module applies a matrix transposition before processing, so that the residual structure operates along the feature dimension $F$. This effectively expands the representation from $\mathbb{R}^{H \times F}$ to $\mathbb{R}^{H \times 2F}$. Despite this reconfiguration, the self-attention mechanism within the module remains effective, now learning dependencies across variables at each temporal position and enabling a richer understanding of cross-feature interactions.

## 2.6 Overall-Architecture of TIM

Having examined the details of each branch in our feature/temporal/residual decomposition paradigm, we now summarize the overall workflow of TIM in Algorithm 2. This unified view illus-

trates how the additive (temporal), interaction (feature), and residual components collaborate within a lightweight, all-MLP framework. The resulting pipeline highlights how principled decomposition, residual correction, and projection are integrated to form a complete forecasting architecture.

---

**Algorithm 2** TIM Overall Workflow

---

**Require:** Input look-back series $X_{\text{in}} \in \mathbb{R}^{L \times F}$, where $L$ is the context length and $F$ the number of variables; prediction horizon $P$; hidden dimension $H$.
**Ensure:** Forecast $Y \in \mathbb{R}^{P \times F}$
    $X \leftarrow \text{Normalize}(X_{\text{in}})$
    $X \leftarrow \text{Transpose}(X) \ \{X \in \mathbb{R}^{F \times L}\}$
    $X \leftarrow \text{TimeEncoder}(X) \ \{X \in \mathbb{R}^{F \times H}\}$
    $X_{\text{time}} \leftarrow \text{TIME FUSION}(X) \ \{\text{additive (main) effects}\}$
    $X_{\text{feat}} \leftarrow \text{Transpose}(\text{FEAT FUSION}(\text{Transpose}(X))) \ \{\text{interaction effects}\}$
    $X_{\text{res}} \leftarrow \text{RES FUSION}(X - X_{\text{time}} - X_{\text{feat}}) \ \{\text{residual correction}\}$
    $Y \leftarrow X_{\text{time}} + X_{\text{feat}} + X_{\text{res}} \ \{Y \in \mathbb{R}^{F \times H}\}$
    $O \leftarrow \text{Proj}(Y) \ \{O \in \mathbb{R}^{F \times P}\}$
    $O \leftarrow \text{Transpose}(O)$
    $\hat{Y} \leftarrow \text{DeNormalize}(O)$
    **return** $\hat{Y} \in \mathbb{R}^{P \times F}$

---

# 3 ANOVA-ALIGNED CROSS-DECOMPOSITION FOR TIM

**Goal.** We align the TIM architecture with the population regression target

$$f_t^\star := \mathbb{E}\big[Y_t \mid B_t, \widetilde{X}_t, \widetilde{Z}_t\big],$$

where $B_t \in \mathbb{R}^{d_B}$ are temporal bases (Fourier/polynomial), $\widetilde{X}_t \in \mathbb{R}^{DL}$ stacks $D$ series over a lookback of length $L$, $\widetilde{Z}_t \in \mathbb{R}^{EL}$ stacks exogenous lags, and $x_t^{(0)} = [B_t^\top, \widetilde{X}_t^\top, \widetilde{Z}_t^\top]^\top$.

## 3.1 POPULATION TARGETS AND THE "NON-ADDITIVE" INTERACTION

Define zero-mean subspaces in $L^2(P_t)$:

$$\mathcal{H}_t^B = \{g(B_t) : \mathbb{E}g = 0\}, \quad \mathcal{H}_t^X = \{h(\widetilde{X}_t) : \mathbb{E}h = 0\}, \quad \mathcal{H}_t^Z = \{q(\widetilde{Z}_t) : \mathbb{E}q = 0\}.$$

Pairwise and triple interaction subspaces are

$$\mathcal{H}_t^{BX} = \{r(B_t, \widetilde{X}_t) : \mathbb{E}[r \mid B_t] = \mathbb{E}[r \mid \widetilde{X}_t] = 0\}, \quad \mathcal{H}_t^{BZ}, \ \mathcal{H}_t^{XZ}, \ \mathcal{H}_t^{BXZ} \text{ analogously.}$$

**Theorem 1 (Time-indexed Hoeffding / ANOVA).** Let $\mu_t = \mathbb{E}[Y_t]$. There exist unique components
$f_t^B \in \mathcal{H}_t^B, \ f_t^X \in \mathcal{H}_t^X, \ f_t^Z \in \mathcal{H}_t^Z, \ f_t^{BX} \in \mathcal{H}_t^{BX}, \ f_t^{BZ} \in \mathcal{H}_t^{BZ}, \ f_t^{XZ} \in \mathcal{H}_t^{XZ}, \ f_t^{BXZ} \in \mathcal{H}_t^{BXZ}$
such that

$$f_t^\star = \mu_t + f_t^B + f_t^X + f_t^Z + f_t^{BX} + f_t^{BZ} + f_t^{XZ} + f_t^{BXZ}, \tag{2}$$

and the components are pairwise orthogonal in $L^2(P_t)$. In particular,

$$f_t^{BX} = \mathbb{E}[Y_t \mid B_t, \widetilde{X}_t] - \mathbb{E}[Y_t \mid B_t] - \mathbb{E}[Y_t \mid \widetilde{X}_t] + \mathbb{E}[Y_t].$$

The orthogonal projection of $f_t^\star$ onto any chosen subset is the unique MSE-optimal approximation.
*Proof:* Appendix B.1.

## 3.2 ANOVA/HOEFFDING IN A NUTSHELL AND RELEVANCE TO MTS

ANOVA/Hoeffding is the orthogonal expansion of a regression function into *main effects* and *interactions* for chosen groups. For $(B_t, \widetilde{X}_t, \widetilde{Z}_t)$, it yields the unique decomposition equation 2. This is apt for multivariate time-series: (i) it separates global temporal structure ($B_t$) and marginal per-variable lag structure ($\widetilde{X}_t, \widetilde{Z}_t$) from cross-group dependencies; (ii) it provides stable, $L^2$-orthogonal attribution; and (iii) it sets a principled target: the cross branch parameterizes the *orthogonal complement* (interactions).

### 3.3 WHY EXPLICITLY MODEL THE *non-additive* PART?

By a Pythagorean identity (Appendix Lemma 1), for any additive predictor $\mu_t + g_t^B + g_t^X + g_t^Z$, the excess risk over the Bayes risk equals the $L^2$ energy of the interaction remainder:

$$\mathbb{E}\Big[\big\|Y_t - (\mu_t + g_t^B + g_t^X + g_t^Z)\big\|^2\Big] - \mathbb{E}\Big[\big\|Y_t - f_t^\star\big\|^2\Big] \;=\; \big\|f_t^{BX} + f_t^{BZ} + f_t^{XZ} + f_t^{BXZ}\big\|_{L^2(P_t)}^2.$$

Thus, if the *non-additive* component is nonzero, ignoring interactions is suboptimal.

**Architectural alignment.** We parameterize *main effects* $f_t^B, f_t^X, f_t^Z$ with lightweight modules, and *interactions* $f_t^{BX}, f_t^{BZ}, f_t^{XZ}, f_t^{BXZ}$ with the cross branch.

### 3.4 CROSS REPRESENTATION ON WINDOWED INPUTS

The DCN-style cross stack at time $t$ is

$$x_t^{(\ell+1)} \;=\; x_t^{(\ell)} \;+\; \mathrm{Diag}(x_t^{(0)})\big(W_\ell x_t^{(\ell)} + b_\ell\big), \qquad \ell = 0, \ldots, k-1. \tag{3}$$

**Theorem 2 (Bounded degree).** Each coordinate of $x_t^{(k)}$ is a multivariate polynomial of total degree at most $k+1$ in $(B_t, \widetilde{X}_t, \widetilde{Z}_t)$. *Proof:* Appendix B.2.

**Theorem 3 (Interaction-only under masks).** If within-group blocks of $W_\ell$ are zero for groups $B$, $\widetilde{X}$, $\widetilde{Z}$, then any degree-$\geq 2$ monomial in $x_t^{(k)}$ involves variables from at least two distinct groups. *Proof:* Appendix B.2.

### 3.5 COEFFICIENT TENSORS AND CONTROLLABLE CAPACITY

For homogeneous parts set $b_\ell = 0$. Fix an output coordinate $j$; let $C_{t,j}^{(d)}$ be the degree-$d$ coefficient tensor.

**Theorem 4 (Explicit coefficient formula).**
$$C_{t,j}^{(d)}[\alpha_1, \ldots, \alpha_d] = \sum_{0 \leq \ell_1 < \cdots < \ell_{d-1} \leq k-1} (W_{\ell_{d-1}})_{j,\alpha_d} \cdots (W_{\ell_1})_{\alpha_2, \alpha_1}.$$

*Proof:* Appendix B.4.

**Corollary 1 (CP-rank bound under low-rank cross maps).** If $W_\ell = U_\ell V_\ell^\top$ with $\mathrm{rank}(W_\ell) \leq r_\ell$, then

$$R_{\mathrm{CP}}(C_{t,j}^{(d)}) \;\leq\; \sum_{\ell_1 < \cdots < \ell_{d-1}} \prod_{m=1}^{d-1} r_{\ell_m} \;\leq\; \binom{k}{d-1} r^{d-1}.$$

*Proof:* Appendix B.4.

### 3.6 RELATION TO TRUNCATED POLYNOMIAL KERNELS

Let $\Phi_{\leq D}(x_t^{(0)})$ list all monomials in $(B_t, \widetilde{X}_t, \widetilde{Z}_t)$ of degree $\leq D$.

**Theorem 5 (Inclusion into degree-$\leq D$ polynomials).** With $D = k+1$, each coordinate of $x_t^{(k)}$ is a linear functional of $\Phi_{\leq D}(x_t^{(0)})$. *Proof:* Appendix B.5.

**Proposition 1 (Strict containment by parameter dimension).** If the number of free parameters $P$ in $\{W_\ell, b_\ell\}$ satisfies $P < M(D)$, then the realized family is a strict subset of all degree-$\leq D$ polynomials. *Proof:* Appendix B.5.

### 3.7 EXPRESSIVITY GAPS WE CLOSE

**Theorem 6 (Additive projection failure for multiplicative truth).** Under $B_t \perp \widetilde{X}_t$ and centering, if $Y_t = \beta\, u(B_t) v(\widetilde{X}_t) + \varepsilon_t$, the additive projection is zero, while a degree-2 cross term represents the signal. *Proof:* Appendix B.6.

**Theorem 7 (Separation-rank lower bound for variable–lag kernels).** For selection-type variable–lag couplings $K$, $\mathrm{sep\text{-}rank}(K) \geq |\mathcal{S}|$. *Proof:* Appendix B.6.

# 4 EXPERIMENTS

## 4.1 DATASET DESCRIPTION

We evaluate our model on eight widely used benchmarks: the ETT datasets with four subsets (ETTh1, ETTh2, ETTm1, ETTm2), as well as Weather, Solar-Energy, Electricity, and Traffic Zhou et al. (2021); Zeng et al. (2023); Hebrail & Berard (2006); Zhao et al. (2019). These datasets provide robust testbeds for assessing long-horizon forecasting performance.

## 4.2 MAIN RESULT

In our experimental setup for model evaluation, we have standardized the parameters across all models to ensure a fair comparison on a uniform platform. Specifically, we have fixed the input dimension to 96 and varied the prediction horizon for time series forecasting, encompassing lengths of [96, 192, 336, 720]. This approach allows for a comprehensive assessment of model performance under different forecasting scenarios. To measure various variables on a consistent scale, we compute the Mean Squared Error (MSE) and Mean Absolute Error (MAE) on the normalized data provided by Revin (Kim et al. (2021)). Additional details regarding the experimental settings, encompassing training specifics and hyperparameters, are furnished in the Appendix. The experiments were implemented using PyTorch (Paszke et al. (2019)) and executed on a single NVIDIA 4090 GPU with 24GB of memory.

For the smaller-scale datasets, such as ETT and Exchange, we have adopted a consistent set of hyperparameters to facilitate a rigorous comparison. Specifically, we have set the number of hidden layers (d_model) to 4, the number of encoder layers (e_layers) to 2, the dropout rate to 0.25, and the learning rate to 1e-3. These configurations have been chosen to balance model complexity and computational efficiency, aiming to achieve optimal performance on the specified datasets.

By adhering to these standardized parameters and experimental protocols, we aim to provide a robust and unbiased evaluation of the different models under investigation, enabling a more meaningful comparison of their strengths and limitations within the context of time series forecasting.

We select 7 SOTA baseline studies. We are focusing on both MLP-based and Transformer-based methods. We added DLinear (Zeng et al. (2023)), RLinear (Li et al. (2023)), TSMixer (Ekambaram et al. (2023)) and TimeMixer (Wang et al. (2024)). We also added PatchTST (Nie et al. (2023)) and iTransformer (Liu et al. (2024)). All SOTAs used Revin (Kim et al. (2021)) as normalization layer.

Results of the main experiments are reported in Table 1 and Table 3. The best outcomes are shown in **red bold**, with the second-best underlined in blue, facilitating direct comparison. Our model consistently outperforms existing state-of-the-art (SOTA) methods, achieving strong results on long-term forecasting and multivariate prediction with a lightweight MLP backbone. These gains stem from our decomposition framework, which separates forecasting signals into additive and non-additive subspaces. The temporal branch captures additive (main) effects, the feature branch models cross-variable interactions, and the residual branch stabilizes optimization. This explicit separation allows both trend-like dynamics and complex dependencies to be effectively represented, explaining the consistent improvements across datasets. The integration of linear self-attention further supports scalable temporal modeling while maintaining efficiency, enabling TIM to balance accuracy and complexity more effectively than prior architectures.

## 4.3 ABLATION STUDY

To verify the effectiveness of each TIM component, we conducted a detailed ablation study on the proposed feature/time/residual decomposition paradigm. The results are reported in Table 4. The prefix "wo" (used as a subscript) denotes "without," indicating the exclusion of specific modules during evaluation. The best results are highlighted in **bold red**, while the second-best are underlined in blue, providing a clear comparison of different configurations.

The ablation results confirm that each branch is indispensable, consistent with our theoretical perspective that forecasting can be decomposed into additive (main) and non-additive (interaction) components with an additional residual correction. Notably, the Time and Res branches share the same architecture but differ in operational sequence and input formulation, namely $Time_{wo}$ and $Res_{wo}$.

Table 1: Multivariate forecasting results with prediction lengths in {96, 192, 336, 720} for eight benchmark datasets and fixed lookback length 96. Results are averaged from all prediction lengths. Avg means further averaged by subsets. Full results are listed in Table 3

| Models (Mean) | TIM Ours | | DLinear 2023 | | PatchTST 2023 | | FreTS 2024 | | RLinear 2023 | | TSMixer 2023 | | TimeMixer 2024 | | iTransFormer 2024 | | TimesNet 2023 | |
|---|---|---|---|---|---|---|---|---|---|---|---|---|---|---|---|---|---|---|
| Metric | mse | mae | mse | mae | mse | mae | mse | mae | mse | mae | mse | mae | mse | mae | mse | mae | mse | mae |
| ETTh1 | **0.434** | **0.433** | 0.452 | 0.447 | 0.440 | 0.442 | 0.464 | 0.447 | 0.443 | **0.431** | 0.456 | 0.446 | 0.465 | 0.450 | 0.448 | 0.443 | 0.531 | 0.491 |
| ETTh2 | 0.377 | 0.402 | 0.526 | 0.498 | 0.379 | 0.405 | 0.448 | 0.457 | 0.385 | 0.407 | 0.396 | 0.414 | **0.368** | **0.398** | 0.382 | 0.407 | 0.429 | 0.434 |
| ETTm1 | **0.382** | **0.397** | 0.404 | 0.408 | 0.444 | 0.457 | 0.432 | 0.438 | 0.409 | 0.400 | 0.401 | 0.406 | 0.403 | 0.411 | 0.404 | 0.406 | 0.620 | 0.580 |
| ETTm2 | **0.272** | **0.318** | 0.337 | 0.388 | 0.281 | 0.328 | 0.284 | 0.328 | 0.287 | 0.328 | 0.290 | 0.332 | 0.298 | 0.338 | 0.291 | 0.334 | 0.333 | 0.351 |
| electricity | **0.172** | **0.268** | 0.210 | 0.296 | 0.223 | 0.2327 | 0.206 | 0.294 | 0.215 | 0.293 | 0.183 | 0.282 | 0.179 | 0.278 | 0.175 | 0.270 | 0.313 | 0.384 |
| solar_AL | 0.244 | 0.271 | 0.327 | 0.397 | 0.244 | 0.349 | 0.268 | 0.322 | 0.356 | 0.350 | 0.257 | 0.292 | 0.268 | 0.298 | 0.239 | 0.280 | **0.197** | **0.244** |
| traffic | 0.469 | 0.292 | 0.626 | 0.386 | 0.500 | **0.287** | 0.556 | 0.365 | 0.624 | 0.375 | 0.510 | 0.348 | 0.506 | 0.335 | **0.462** | 0.307 | 0.640 | 0.348 |
| weather | **0.241** | **0.270** | 0.266 | 0.318 | 0.248 | 0.275 | 0.249 | 0.278 | 0.269 | 0.288 | 0.246 | 0.276 | 0.261 | 0.284 | 0.252 | 0.277 | 0.273 | 0.291 |
| 1st count | **5** | **4** | 0 | 0 | 0 | 1 | 0 | 0 | 0 | 1 | 0 | 0 | 1 | 1 | 1 | 0 | 1 | 1 |

In *Time*$_{wo}$, the model learns temporal transitions across transposed multivariate slices, whereas in *Res*$_{wo}$, it processes univariate time series as tokens to capture cross-variable dependencies.

The Feat branch, aligned with the interaction component, embeds multiple variables into each temporal token, allowing the model to capture delayed events and heterogeneous physical measurements. However, without the complementary additive signal, this design risks entangling variate-specific patterns too early, sometimes yielding ineffective attention maps.

Conversely, the Time branch focuses on the additive structure, embedding time points of individual series into variate tokens to strengthen multivariate correlations. This explains why the *Res*$_{wo}$ setting still achieves performance close to the full model—its residual signal partly compensates for missing additive effects—yet remains inferior to the complete decomposition.

Finally, while prior work has cautioned against dataset-specific tailoring due to overfitting risks Li et al. (2024), our ablation experiments demonstrate that across most benchmarks the full TIM consistently achieves the best performance. This validates our principled decomposition: each branch contributes uniquely to disentangling additive, interaction, and residual components, and together they yield a unified, stable, and theoretically grounded architecture for long-sequence forecasting.

## 4.4 MODEL EFFICIENCY

We compare the training memory consumption and runtime of TIM against recent state-of-the-art models. Our results consistently show that TIM achieves superior efficiency in both GPU memory utilization and execution time, benefiting from its all-MLP backbone and linear-time propagation. Figure 2 illustrates the trade-off between model size and predictive accuracy: the horizontal axis denotes Mean Squared Error (MSE), the vertical axis represents the logarithm of parameter count, and marker size indicates floating-point operations (FLOPs). Despite having a parameter count comparable to other advanced approaches, TIM achieves substantially lower MSE across prediction lengths. This advantage arises from its decomposition-based architecture: the temporal branch captures additive effects efficiently, the feature branch models non-additive interactions beyond standard MLPs, and the residual branch stabilizes optimization. As a result, TIM not only avoids the quadratic overhead of Transformer attention but also surpasses conventional MLP baselines in representational capacity. Together, these properties enable TIM to balance efficiency and expressivity more effectively than existing architectures, achieving strong predictive accuracy with a streamlined, resource-efficient design. To further substantiate this advantage, we compare the asymptotic time and memory complexities of different architectures in Table 2. As shown, Transformers incur quadratic cost in sequence length $L$, while MLP baselines scale poorly with hidden dimension $H$. In contrast, TIM attains linear complexity in both $L$ and the number of variables $F$, while explicitly modeling additive and interaction effects. This demonstrates that TIM combines the scalability of linear models with the expressivity of cross-feature architectures, providing a principled and efficient solution for long-sequence forecasting.

Table 2: Asymptotic training complexity with batch size $B$, sequence length $L$, hidden dimension $H$, #variables $F$, DCN depth $k$, and cross-map rank $r$.

| Model | Time Complexity | Memory Complexity |
|---|---|---|
| Transformer (self-attention) | $\mathcal{O}(B\,L^2 H)$ | $\mathcal{O}(B\,L^2)$ |
| Linear Transformer | $\mathcal{O}(B\,LH)$ | $\mathcal{O}(B\,LH)$ |
| Standard MLP | $\mathcal{O}(B\,L\,H^2)$ | $\mathcal{O}(B\,L\,H)$ |
| TIM (ours) | $\mathcal{O}\big(B\,(L+k\,F\,r)\,H\big)$ | $\mathcal{O}\big(B\,(L+F+k\,r)\,H\big)$ |

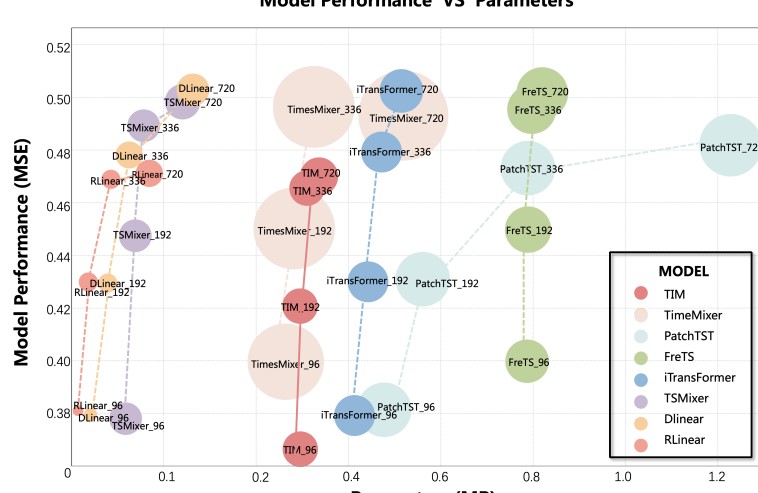

Figure 2: **Parameters** vs **Model performance (MSE)**. We reported the experiment This figure presents the experimental results for our models across various prediction lengths (pred_len) on the ETTh1 dataset. Notably, our all-MLP TIM has achieved SOTA performance while possessing a significantly smaller number of parameters compared to transformer-based models. The horizontal axis represents the logarithmic scale of model parameters (MB), and the vertical axis indicates the model performance measured by Mean Squared Error (MSE). For clarity in presentation, we applied a square root transformation to the model's parameter size, expressed in megabytes (MB).

## 5 CONCLUSION AND FUTURE WORK

In this paper, we presented TIM, a lightweight forecaster that achieves state-of-the-art performance in long-sequence time series forecasting while maintaining low computational and memory complexity. Through a principled feature/time/residual decomposition, TIM explicitly models both additive and interaction effects with minimal overhead, making it particularly suitable for resource-constrained environments.

Our experiments across diverse benchmarks demonstrate that this decomposition not only improves predictive accuracy but also enhances stability and interpretability. In particular, our theoretical risk analysis and empirical ablations reveal that a substantial portion of the forecasting signal lies in the non-additive interaction subspace. By aligning the architecture with this subspace and controlling interaction order and rank, TIM achieves consistent accuracy gains together with clear, component-wise attribution.

While TIM demonstrates strong efficiency and accuracy, further improvements are needed to capture higher-order dependencies and adapt to non-stationary dynamics. Future work will focus on enriching the modeling of the interaction subspace and developing adaptive mechanisms that preserve efficiency while extending expressivity. By advancing along these directions, we aim to further strengthen the predictive power and practical applicability of TIM in real-world forecasting scenarios.

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

## A APPENDIX

## B PROOFS FOR SECTION 3

### B.1 PROOFS OF ANOVA/HOEFFDING STATEMENTS

**Proof of Theorem 1.** Fix $t$. Work in $L^2(P_t)$ with inner product $\langle U, V \rangle = \mathbb{E}[UV]$. Let $u := f_t^\star - \mu_t$. Define the projection operators $\Pi_B u = \mathbb{E}[u \mid B_t] - \mathbb{E}[u]$, $\Pi_X u = \mathbb{E}[u \mid \widetilde{X}_t] - \mathbb{E}[u]$, $\Pi_Z u = \mathbb{E}[u \mid \widetilde{Z}_t] - \mathbb{E}[u]$. Set $f_t^B = \Pi_B u$, $f_t^X = \Pi_X u$, $f_t^Z = \Pi_Z u$, and $f_t^{BX} = \mathbb{E}[u \mid B_t, \widetilde{X}_t] - f_t^B - f_t^X$, $f_t^{BZ} = \mathbb{E}[u \mid B_t, \widetilde{Z}_t] - f_t^B - f_t^Z$, $f_t^{XZ} = \mathbb{E}[u \mid \widetilde{X}_t, \widetilde{Z}_t] - f_t^X - f_t^Z$, $f_t^{BXZ} = u - f_t^B - f_t^X - f_t^Z - f_t^{BX} - f_t^{BZ} - f_t^{XZ}$. These satisfy the required conditional-centering constraints and are pairwise orthogonal by the tower property. Uniqueness and MSE optimality follow from orthogonal projection. □

### B.2 PROOFS FOR CROSS DEGREE AND MASKS

**Lemma (one-step degree growth).** If each coordinate of $u$ is a polynomial of degree $\leq D$, then $\mathrm{Diag}(x_t^{(0)})(Wu + b)$ has degree $\leq D+1$. **Proof.** $Wu$ is degree $\leq D$; multiplication by $\mathrm{Diag}(x_t^{(0)})$ adds one; $\mathrm{Diag}(x_t^{(0)})b$ is degree 1. □

**Proof of Theorem 2.** Induct on $\ell$ in equation 3 with base $x_t^{(0)}$ (degree 1); apply the lemma at each step. □

**Proof of Theorem 3.** Any degree-$\geq 2$ term includes a factor $\mathrm{Diag}(x_t^{(0)})W_\ell u$. Fix an output coordinate $j$ in group $G \in \{B, \widetilde{X}, \widetilde{Z}\}$. Since $W_\ell^{GG} = 0$, the contributing coordinate of $u$ lies in a different group $G' \neq G$. Multiplication by the $j$-th entry of $x_t^{(0)}$ injects a variable from $G$, while $u$ contains a variable from $G'$, producing a cross-group monomial. Composition preserves this property. □

### B.3 LINEAR ATTENTION PRESERVES THE CROSS-DETERMINED POLYNOMIAL DEGREE

**Proposition 1 (Degree preservation of axis-wise linear attention).** Let $u_s(x^{(0)}) \in \mathbb{R}^H$ denote the output of the DCN-style cross stack at position $s$, where each coordinate of $u_s$ is a polynomial in $x^{(0)} = [B_t^\top, \widetilde{X}_t^\top, \widetilde{Z}_t^\top]^\top$ of total degree at most $D$ (hence bounded by the cross depth). Consider

axis-wise linear attention with feature map $\phi : \mathbb{R}^{d_h} \to \mathbb{R}^r$ and define

$$y_t = \frac{\phi(q_t)^\top \left( \sum_s \phi(k_s) v_s^\top \right)}{\phi(q_t)^\top \left( \sum_s \phi(k_s) \right)} \quad \text{with} \quad v_s = W_v u_s.$$

Assume the following:

(A1) (*Value linearity*) $v_s = W_v u_s$ is linear in $u_s$; thus $\deg(v_s) \leq D$.

(A2) (*No multiplicative feedback*) The attention output $y_t$ is used *additively* outside the cross stack, without multiplicative gating or feedback into the cross recursion.

(A3) (*Group separation*) Queries/keys are affine maps of variables from a group $G$ disjoint from the cross pathway inside DCN (e.g., a main-effect stream): $q_t = A_q z_t + a_q$, $k_s = A_k z_s + a_k$ with $z_\cdot \in \sigma(G)$. Within DCN, group-wise zero-block masks are enforced so that $z_\cdot$ does not multiplicatively couple inside the cross recursion.

(A4) (*PSD kernel smoothing with $u$-independent normalizer*) $\phi$ is a fixed *linear* (or affine) positive feature map, and the normalizer $c_t := \phi(q_t)^\top \sum_s \phi(k_s)$ depends only on $z_\cdot$ but not on $\{u_s\}$, ensuring that $c_t^{-1}$ does not turn $\{u_s\}$ into rational functions.

Then each coordinate of $y_t$ is an *affine* functional of $\{u_s\}_s$; in particular, the total polynomial degree of $y_t$ in $x^{(0)}$ is at most $D$.

*Proof.* From (A1), $v_s$ is linear in $\{u_s\}$ and $\deg(v_s) \leq D$. From (A3)–(A4), both $M_t := \phi(q_t)^\top \sum_s \phi(k_s)$ and the normalizer $c_t$ depend only on $z_\cdot$ and are independent of $\{u_s\}$. Hence

$$y_t = c_t^{-1} \phi(q_t)^\top \left( \sum_s \phi(k_s) W_v u_s^\top \right)$$

is a linear functional of $\{u_s\}$. A linear operator cannot increase the maximum polynomial degree of its inputs, so $\deg(y_t) \leq D$. By (A2), this linear transformation is not fed back into the cross recursion, and therefore does not trigger a new multiplicative chain in the DCN cross stack. $\qquad\square$

**Corollary 1 (Mask-preserving property under linear attention).** Under the assumptions of Proposition 1 and the within-group zero-block masks for the cross matrices $\{W_\ell\}$, any degree-$\geq 2$ monomial that appears in $y_t$ must already appear in some $u_s$. In particular, axis-wise linear attention does not synthesize new within-group higher-order monomials and preserves the "interaction-only" nature enforced by the masks.

*Proof.* By Proposition 1, $y_t$ is a linear combination of $\{u_s\}$ with coefficients depending only on $z_\cdot$, so there is no multiplicative coupling with $\{u_s\}$. Thus the monomial set of $y_t$ is contained in the linear span of the monomial sets of $\{u_s\}$, and no new higher-order terms can be created. Mask constraints guarantee that all degree-$\geq 2$ monomials in $\{u_s\}$ are cross-group interactions, and linear combinations preserve this property. $\qquad\square$

**Discussion (when the bound can fail).** If (A3) or (A4) is violated—for instance, if $\phi$ depends nonlinearly on $\{u_s\}$ or if the normalizer $c_t$ involves $u$—then $y_t$ may become a *rational function* of $\{u_s\}$ and introduce higher-order terms. If (A2) is violated (e.g., feeding the attention output multiplicatively back into the cross recursion), then the total polynomial degree increases by at least one. These conditions delineate the "safe zone" where linear attention acts as PSD kernel smoothing plus group separation with additive usage, in which case it only *transports* features linearly without raising the maximum polynomial degree determined by the DCN cross stack.

## B.4 Proofs for coefficient tensors and CP-rank

**Proof of Theorem 4.** Write $T_\ell(u) = \text{Diag}(x_t^{(0)})(W_\ell u + b_\ell)$ and expand $x_t^{(k)} = \left( \prod_{\ell=k-1}^0 (I + T_\ell) \right) x_t^{(0)}$. Retaining degree-$d$ terms corresponds to $m = d-1$ occurrences of $W_\ell$ and one multiplication by a coordinate of $x_t^{(0)}$, yielding the chain formula. $\qquad\square$

**Corollary 1 (CP-rank bound).** Assume $W_\ell = U_\ell V_\ell^\top$ with rank at most $r_\ell$. For a fixed chain, the contribution is a sum of $\prod_m r_{\ell_m}$ rank-1 outer products across the $d$ index modes. Summing over all chains gives the bound; the looser bound uses $r = \max_\ell r_\ell$. $\qquad\square$

### B.5 Proofs for polynomial-kernel relations

**Proof of Theorem 5.** By Theorem 2, $x_t^{(k)}$ has degree $\leq D = k+1$ in $(B_t, \widetilde{X}_t, \widetilde{Z}_t)$; hence each coordinate is a linear combination of monomials in $\Phi_{\leq D}(x_t^{(0)})$. $\square$

**Proof of Proposition 1.** Let $V$ be the $M(D)$-dimensional space spanned by $\Phi_{\leq D}$. The parameterization map has dimension at most $P$; when $P < M(D)$ it cannot cover an open subset of $V$, so the family is proper. $\square$

### B.6 Proofs for expressivity gaps

**Proof of Theorem 6.** Assume $B_t \perp \widetilde{X}_t$ and $\mathbb{E}[u(B_t)] = \mathbb{E}[v(\widetilde{X}_t)] = 0$. For any $g(B_t)$, $\langle Y_t, g(B_t) \rangle = \beta \, \mathbb{E}[u(B_t)g(B_t)] \, \mathbb{E}[v(\widetilde{X}_t)] = 0$; similarly for $h(\widetilde{X}_t)$. Therefore the $L^2$ projection onto $\{g + h\}$ is zero. A degree-2 cross monomial containing $u(B_t)v(\widetilde{X}_t)$ represents the signal. $\square$

**Proof of Theorem 7.** Index $\widetilde{X}_t$ by $(i, \tau)$ with $i \in \{1, \ldots, D\}$ and $\tau \in \{0, \ldots, L-1\}$. Let $E_{ij}$ be the $D \times D$ unit matrix at $(i, j)$ and $T_\Delta$ the $L \times L$ Toeplitz lag-$\Delta$ indicator. Then $K = \sum_{(i,j,\Delta) \in \mathcal{S}} \alpha_{ij\Delta}(E_{ij} \otimes T_\Delta)$. Vectorizing gives $\sum c_{ij\Delta} \operatorname{vec}(E_{ij}) \operatorname{vec}(T_\Delta)^\top = 0$ only if $c_{ij\Delta} = 0$ for all indices. Thus sep-rank$(K) \geq |\mathcal{S}|$. $\square$

### B.7 Risk decomposition identities (why interactions matter)

**Lemma 1 (Pythagorean identity).** Let $f_t^\star = \mu_t + f_t^B + f_t^X + f_t^Z + f_t^{BX} + f_t^{BZ} + f_t^{XZ} + f_t^{BXZ}$ be the orthogonal decomposition. Then for any $g_t^B \in \mathcal{H}_t^B$, $g_t^X \in \mathcal{H}_t^X$, $g_t^Z \in \mathcal{H}_t^Z$,

$$\mathbb{E}\Big[\big\|Y_t - (\mu_t + g_t^B + g_t^X + g_t^Z)\big\|^2\Big] = \mathbb{E}[\|Y_t - f_t^\star\|^2] + \|f_t^B - g_t^B\|_2^2 + \|f_t^X - g_t^X\|_2^2 + \|f_t^Z - g_t^Z\|_2^2 + \|f_t^{BX} + f_t^{BZ} + f_t^{XZ} + f_t^{BXZ}\|_2^2.$$

**Theorem 1 (Additive risk gap equals interaction energy).** The additive model that minimizes the LHS equals $\mu_t + f_t^B + f_t^X + f_t^Z$; its excess risk over the Bayes risk equals $\|f_t^{BX} + f_t^{BZ} + f_t^{XZ} + f_t^{BXZ}\|_{L^2(P_t)}^2$. $\square$

## C Related Work

### C.1 Problem Statement

In the context of multivariate time series analysis, let $X = \{x_1^{(c)}, \ldots, x_L^{(c)}\}_{f=1}^F$ denote a collection of $F$ feature channels, where each channel $c$ comprises an independent sequence of $L$ observations within a look-back window. The channel index $f$ will be omitted in subsequent discussions for simplicity. The objective of the forecasting task is to predict the future values of the time series over the next $pred\_len$ time steps, denoted as $\hat{X}_{L+1:L+P}$, based on the historical data $X_{1:L}$, where pred_len is abbreviated as $P$. This prediction is achieved through a forecasting function $F(\cdot)$, which is instantiated as an MLP-based model in this study. Our primary goal is to mitigate the high computational cost and performance degradation associated with long-term data and to enhance model prediction capabilities through multivariable feature interaction and long-term series distribution migration modelling. This approach seeks to improve the forecasting outcome $X'$, specifically by minimizing the error between the predicted values $X'$ (i.e., $F(X_{1:L})$) and the true future values $\hat{X}_{L+1:L+P}$. Traditionally, time series data are usually subjected to batch normalization before being input into prediction models. However, recent research has highlighted the efficacy of utilizing a reversible instance normalization (RevIN: Kim et al. (2022)) in addressing the challenges posed by distribution shifts in time-series forecasting problems.

### C.2 Temporal Modeling for LSTF

In the realm of Long Short-Term Forecasting (LSTF) tasks, Transformer-based and MLP-based models have emerged as the preeminent backbones due to their exceptional temporal modelling capabilities. Deviating from the Vanilla Transformer (Ashish (2017)), recent research has advanced the field significantly. Notably, Informer (Zhou et al. (2021)) introduced an innovative strategy

whereby timestamps are encoded as supplementary positional encodings through the deployment of learnable embedding layers. This advancement, along with subsequent works such as Autoformer (Wu et al. (2021)) and FEDformer (Zhou et al. (2022)), has firmly established these foundational architectures as widely acknowledged solutions for addressing LSTF challenges. Subsequent endeavours have introduced iTransformer, a variant that ingeniously applies the attention mechanism and feed-forward network on inverted dimensions. This innovation not only diversifies the Transformer family but also propels its performance to new heights, further demonstrating the potential and adaptability of Transformer-based models in handling complex tasks. Furthermore, the MLPs (Oreshkin et al. (2019); Challu et al. (2023)) achieve favourable performance in both forecasting performance and efficiency for LSTF tasks. Previous research has demonstrated that MLPs can achieve the same top level of performance as Transformers in long-term sequential forecasting tasks using trend season decomposition methods (Zeng et al. (2023)). Recent research on TimeMixer (Wang et al. (2024)) has elegantly capitalized on disentangled multiscale series, leveraging them effectively in both the past extraction and future prediction phases. This approach has demonstrated remarkable achievements, consistently attaining state-of-the-art performances across both long-term and short-term forecasting tasks, while also exhibiting favourable run-time efficiency, underscoring its practical significance and efficiency in real-world applications.

Traditional sequential models, such as Recurrent Neural Networks (RNNs), frequently encounter issues of gradient vanishing or gradient explosion when dealing with long time series, rendering them challenged in capturing long-range dependencies. The Attention mechanism, by directly computing the relevance between any two positions within the sequence, can mitigate this problem to some extent, enabling the model to process long sequence data more effectively. By incorporating the Attention mechanism, the model is able to dynamically allocate more importance or "focus" on the most relevant parts of the input sequence, regardless of their positions within the sequence. The following equation can formulate the classic attention mechanism, particularly within the framework of self-attention or transformer-based models, Q typically represents the "Query", K denotes the "Key", and V stands for the "Value". we ignore the normalization term for simplicity.

$$\text{Attention}(Q, K, V) = \text{softmax}(QK^T)V \tag{4}$$

In classical attention mechanisms, both spatial and temporal complexities scale with $O(n^2)$, where n represents the sequence length. Consequently, as n increases significantly, the computational burden on Transformer models becomes prohibitively high. Recently, extensive research has focused on addressing this issue by reducing the computational cost of Transformer models. These efforts include various techniques such as Sparse Attention (Wu et al. (2020); Zhang et al. (2024)), and quantization. Additionally, modifications to the attention architecture have been explored to reduce its complexity to $O(n \log(n))$ or even $O(n)$, thereby improving the scalability and efficiency of Transformer models for processing longer sequences.

## C.3 LINEAR ATTENTION

The Attention mechanism of equation 4 can be rewritten in the following way:

$$\text{Attention}(Q, K, V)_i = \frac{\sum_{j=1}^{n} \exp\left(q_i^\top k_j\right) v_j}{\sum_{j=1}^{n} \exp\left(q_i^\top k_j\right)} = \frac{\sum_{j=1}^{n} \text{sim}(q_i, k_j) v_j}{\sum_{j=1}^{n} \text{sim}(q_i, k_j)} \tag{5}$$

Previous research (Wang et al. (2018)) had pointed out that if we use $\text{sim}(q_i, k_j) = \phi(q_i)^\top \varphi(k_j)$ to simplify the calculation of attention, then the complexity problem of attention mechanism should be mitigated. $\phi(x), \varphi(x)$ are defined as $\phi(x) = \varphi(x) = elu(x) + 1$, where $elu(x)$ denotes the Exponential Linear Unit (as introduced by Clevert (2015)). The additional "+1" term ensures that the similarity term remains positive. From the perspective of the result, equation 5 expresses that the core logic of the attention mechanism lies in focusing on everything and the key points. It can be seen from the weighted sum expression of the Attention formula that the self-attention mechanism can help to model the entire time series and automatically help the model focus on the local feature.

In our work, we harness the merits of the linear attention mechanism to explicitly model the multi-variable interaction across the entire time series of individual variables, as well as the evolving features within cross-sectional multi-variable data. This approach endows our model with several advantageous characteristics, including reduced computational complexity, minimized storage requirements, the capability to model the global time series, localized feature attention, and the profi-

ciency to handle multi-variable relationships. We will delve deeper into the intricate architecture of our model in the subsequent method section.

## C.4 FEATURE FUSION

To leverage linear attention effectively in capturing both the multi-variable interactions across the entire time series of individual variables and the evolving features within cross-sectional multi-variable data, our approach aims to extract meaningful global information from the time series and accurately represent the intricate multi-variable relationships. This process is non-trivial and frequently necessitates intricate manual feature engineering or an exhaustive search procedure. Previous work Wang et al. (2017) introduces a novel cross-network that is more efficient in learning certain bounded-degree feature interactions when it keeps the benefits of MLPs without extra complexity. This enables our model to comprehensively analyze and understand the dynamics within and across variables over time.

## C.5 EXPERIMENT SETTING

To ensure a fair comparison across all models on a uniform platform (the time-series-library), we have standardized the parameters. Specifically, we have fixed the input dimension at 96 and varied the prediction horizon for time series forecasting, with lengths including 96, 192, 336, and 720. The batch size was set to 32, the learning rate to 1e-3, the model dimension ($d\_model$) to 512, and the dropout rate to 0.1.

## C.6 MAIN RESULT

Table 3: Multivariate forecasting results with prediction lengths in {96, 192, 336, 720} for eight benchmark datasets and fixed lookback length 96. Our proposed model, ComplexFormer, has achieved state-of-the-art (SOTA) performance across 25 tasks based on the Mean Squared Error (MSE) metric, and on 20 tasks when evaluated using the Mean Absolute Error (MAE) metric. While TIM demonstrates strong performance across various benchmarks, primarily due to its low computational complexity and cross-layer architecture, further improvements can be made to enhance its ability to capture intricate multivariate relationships, particularly in datasets with a large number of variables and extended time series.

| Models | | TIM Ours | | DLinear 2023 | | PatchTST 2023 | | FreTS 2024 | | RLinear 2023 | | TSMixer 2023 | | TimeMixer 2024 | | iTransFormer 2024 | | TimesNet 2023 | |
|---|---|---|---|---|---|---|---|---|---|---|---|---|---|---|---|---|---|---|---|
| Metric | | mse | mae | mse | mae | mse | mae | mse | mae | mse | mae | mse | mae | mse | mae | mse | mae | mse | mae |
| ETTh1 | 96 | 0.367 | 0.391 | 0.386 | 0.399 | 0.383 | 0.402 | 0.400 | 0.409 | 0.385 | 0.393 | 0.384 | 0.403 | 0.408 | 0.413 | 0.384 | 0.403 | 0.408 | 0.426 |
| | 192 | 0.424 | 0.425 | 0.434 | 0.428 | 0.435 | 0.431 | 0.455 | 0.440 | 0.436 | 0.422 | 0.444 | 0.435 | 0.457 | 0.442 | 0.434 | 0.431 | 0.496 | 0.475 |
| | 336 | 0.472 | 0.446 | 0.482 | 0.460 | 0.470 | 0.452 | 0.496 | 0.460 | 0.476 | 0.442 | 0.491 | 0.460 | 0.505 | 0.467 | 0.482 | 0.457 | 0.512 | 0.484 |
| | 720 | 0.471 | 0.469 | 0.504 | 0.502 | 0.479 | 0.476 | 0.506 | 0.481 | 0.478 | 0.467 | 0.505 | 0.485 | 0.492 | 0.478 | 0.491 | 0.482 | 0.708 | 0.580 |
| | AVG | 0.434 | 0.433 | 0.452 | 0.447 | 0.440 | 0.442 | 0.464 | 0.447 | 0.443 | 0.431 | 0.456 | 0.446 | 0.465 | 0.450 | 0.448 | 0.443 | 0.531 | 0.491 |
| ETTh2 | 96 | 0.289 | 0.342 | 0.329 | 0.384 | 0.292 | 0.344 | 0.298 | 0.348 | 0.290 | 0.340 | 0.304 | 0.353 | 0.293 | 0.342 | 0.302 | 0.352 | 0.343 | 0.378 |
| | 192 | 0.374 | 0.393 | 0.435 | 0.448 | 0.373 | 0.395 | 0.382 | 0.399 | 0.378 | 0.395 | 0.402 | 0.409 | 0.375 | 0.394 | 0.379 | 0.399 | 0.449 | 0.432 |
| | 336 | 0.419 | 0.430 | 0.563 | 0.526 | 0.417 | 0.431 | 0.426 | 0.436 | 0.430 | 0.439 | 0.444 | 0.445 | 0.398 | 0.424 | 0.418 | 0.430 | 0.468 | 0.459 |
| | 720 | 0.427 | 0.444 | 0.775 | 0.634 | 0.434 | 0.450 | 0.448 | 0.457 | 0.442 | 0.453 | 0.436 | 0.450 | 0.406 | 0.432 | 0.427 | 0.447 | 0.457 | 0.467 |
| | AVG | 0.377 | 0.402 | 0.526 | 0.498 | 0.379 | 0.405 | 0.448 | 0.457 | 0.385 | 0.407 | 0.396 | 0.414 | 0.368 | 0.398 | 0.382 | 0.407 | 0.429 | 0.434 |
| ETTm1 | 96 | 0.315 | 0.357 | 0.345 | 0.371 | 0.377 | 0.424 | 0.358 | 0.394 | 0.350 | 0.368 | 0.321 | 0.361 | 0.333 | 0.368 | 0.337 | 0.371 | 0.429 | 0.454 |
| | 192 | 0.361 | 0.383 | 0.383 | 0.394 | 0.417 | 0.439 | 0.399 | 0.411 | 0.388 | 0.386 | 0.370 | 0.388 | 0.376 | 0.393 | 0.376 | 0.388 | 0.593 | 0.572 |
| | 336 | 0.386 | 0.402 | 0.414 | 0.414 | 0.465 | 0.466 | 0.433 | 0.439 | 0.419 | 0.406 | 0.415 | 0.414 | 0.408 | 0.418 | 0.423 | 0.414 | 0.679 | 0.601 |
| | 720 | 0.469 | 0.446 | 0.474 | 0.453 | 0.517 | 0.501 | 0.538 | 0.509 | 0.480 | 0.440 | 0.497 | 0.461 | 0.493 | 0.464 | 0.480 | 0.449 | 0.780 | 0.692 |
| | AVG | 0.382 | 0.397 | 0.404 | 0.408 | 0.444 | 0.457 | 0.432 | 0.438 | 0.409 | 0.400 | 0.401 | 0.406 | 0.403 | 0.411 | 0.404 | 0.406 | 0.620 | 0.580 |
| ETTm2 | 96 | 0.172 | 0.253 | 0.186 | 0.282 | 0.176 | 0.261 | 0.180 | 0.262 | 0.182 | 0.265 | 0.183 | 0.267 | 0.183 | 0.267 | 0.184 | 0.268 | 0.188 | 0.268 |
| | 192 | 0.233 | 0.294 | 0.270 | 0.347 | 0.242 | 0.305 | 0.247 | 0.306 | 0.247 | 0.306 | 0.249 | 0.309 | 0.260 | 0.318 | 0.252 | 0.312 | 0.288 | 0.324 |
| | 336 | 0.292 | 0.333 | 0.362 | 0.414 | 0.303 | 0.344 | 0.304 | 0.342 | 0.309 | 0.344 | 0.310 | 0.347 | 0.309 | 0.346 | 0.317 | 0.352 | 0.343 | 0.360 |
| | 720 | 0.391 | 0.392 | 0.527 | 0.507 | 0.402 | 0.402 | 0.406 | 0.402 | 0.408 | 0.400 | 0.417 | 0.407 | 0.438 | 0.420 | 0.411 | 0.405 | 0.512 | 0.450 |
| | AVG | 0.272 | 0.318 | 0.337 | 0.388 | 0.281 | 0.328 | 0.284 | 0.328 | 0.287 | 0.328 | 0.290 | 0.332 | 0.298 | 0.338 | 0.291 | 0.334 | 0.333 | 0.351 |
| electricity | 96 | 0.144 | 0.241 | 0.195 | 0.277 | 0.206 | 0.309 | 0.184 | 0.271 | 0.198 | 0.274 | 0.156 | 0.258 | 0.153 | 0.253 | 0.146 | 0.244 | 0.351 | 0.405 |
| | 192 | 0.164 | 0.259 | 0.194 | 0.280 | 0.214 | 0.321 | 0.188 | 0.277 | 0.198 | 0.277 | 0.174 | 0.274 | 0.169 | 0.270 | 0.162 | 0.256 | 0.293 | 0.375 |
| | 336 | 0.173 | 0.271 | 0.208 | 0.297 | 0.218 | 0.325 | 0.203 | 0.294 | 0.212 | 0.293 | 0.187 | 0.288 | 0.184 | 0.285 | 0.180 | 0.274 | 0.290 | 0.373 |
| | 720 | 0.205 | 0.301 | 0.243 | 0.330 | 0.254 | 0.352 | 0.248 | 0.335 | 0.254 | 0.326 | 0.216 | 0.309 | 0.209 | 0.305 | 0.213 | 0.305 | 0.317 | 0.383 |
| | AVG | 0.172 | 0.268 | 0.210 | 0.296 | 0.223 | 0.2327 | 0.206 | 0.294 | 0.215 | 0.293 | 0.183 | 0.282 | 0.179 | 0.278 | 0.175 | 0.270 | 0.313 | 0.384 |
| solar_AL | 96 | 0.213 | 0.241 | 0.285 | 0.372 | 0.223 | 0.328 | 0.250 | 0.308 | 0.305 | 0.329 | 0.214 | 0.264 | 0.234 | 0.279 | 0.203 | 0.256 | 0.189 | 0.257 |
| | 192 | 0.234 | 0.266 | 0.316 | 0.393 | 0.246 | 0.353 | 0.268 | 0.328 | 0.344 | 0.348 | 0.257 | 0.292 | 0.277 | 0.306 | 0.233 | 0.271 | 0.193 | 0.234 |
| | 336 | 0.261 | 0.287 | 0.352 | 0.413 | 0.260 | 0.365 | 0.285 | 0.336 | 0.386 | 0.364 | 0.280 | 0.307 | 0.284 | 0.307 | 0.266 | 0.304 | 0.200 | 0.238 |
| | 720 | 0.267 | 0.289 | 0.355 | 0.411 | 0.246 | 0.350 | 0.269 | 0.315 | 0.389 | 0.358 | 0.278 | 0.304 | 0.278 | 0.300 | 0.254 | 0.286 | 0.207 | 0.248 |
| | AVG | 0.244 | 0.271 | 0.327 | 0.397 | 0.244 | 0.349 | 0.268 | 0.322 | 0.356 | 0.350 | 0.257 | 0.292 | 0.268 | 0.298 | 0.239 | 0.280 | 0.197 | 0.244 |
| traffic | 96 | 0.447 | 0.277 | 0.650 | 0.398 | 0.475 | 0.277 | 0.542 | 0.357 | 0.646 | 0.386 | 0.487 | 0.338 | 0.472 | 0.316 | 0.427 | 0.299 | 0.593 | 0.333 |
| | 192 | 0.458 | 0.287 | 0.599 | 0.371 | 0.489 | 0.278 | 0.537 | 0.358 | 0.599 | 0.362 | 0.496 | 0.338 | 0.494 | 0.328 | 0.451 | 0.302 | 0.631 | 0.349 |
| | 336 | 0.471 | 0.292 | 0.607 | 0.375 | 0.500 | 0.291 | 0.553 | 0.363 | 0.607 | 0.366 | 0.514 | 0.349 | 0.518 | 0.347 | 0.464 | 0.304 | 0.664 | 0.353 |
| | 720 | 0.503 | 0.310 | 0.648 | 0.398 | 0.535 | 0.302 | 0.590 | 0.380 | 0.645 | 0.385 | 0.541 | 0.368 | 0.540 | 0.350 | 0.506 | 0.324 | 0.673 | 0.359 |
| | AVG | 0.469 | 0.292 | 0.626 | 0.386 | 0.500 | 0.287 | 0.556 | 0.365 | 0.624 | 0.375 | 0.510 | 0.348 | 0.506 | 0.335 | 0.462 | 0.307 | 0.640 | 0.348 |
| weather | 96 | 0.155 | 0.200 | 0.196 | 0.255 | 0.165 | 0.211 | 0.167 | 0.213 | 0.193 | 0.232 | 0.159 | 0.208 | 0.160 | 0.207 | 0.168 | 0.211 | 0.194 | 0.233 |
| | 192 | 0.204 | 0.246 | 0.238 | 0.297 | 0.212 | 0.253 | 0.241 | 0.272 | 0.236 | 0.268 | 0.214 | 0.254 | 0.226 | 0.265 | 0.214 | 0.254 | 0.240 | 0.270 |
| | 336 | 0.262 | 0.289 | 0.283 | 0.333 | 0.268 | 0.292 | 0.269 | 0.295 | 0.288 | 0.304 | 0.273 | 0.294 | 0.286 | 0.307 | 0.273 | 0.296 | 0.292 | 0.307 |
| | 720 | 0.345 | 0.344 | 0.348 | 0.385 | 0.346 | 0.344 | 0.346 | 0.346 | 0.359 | 0.350 | 0.349 | 0.348 | 0.372 | 0.358 | 0.351 | 0.347 | 0.364 | 0.353 |
| | AVG | 0.242 | 0.270 | 0.266 | 0.318 | 0.248 | 0.275 | 0.249 | 0.278 | 0.269 | 0.288 | 0.246 | 0.276 | 0.261 | 0.284 | 0.252 | 0.277 | 0.273 | 0.291 |
| 1st count | | 25 | 21 | 0 | 0 | 2 | 5 | 0 | 0 | 0 | 6 | 0 | 0 | 3 | 3 | 5 | 1 | 5 | 4 |

## C.7 CODE OF ETHICS

We have read and understood the ICLR Code of Ethics, as outlined on the conference website. We fully acknowledge the importance of adhering to these ethical guidelines throughout all aspects of my participation in ICLR, including paper submission, reviewing, and discussions.

Table 4: Ablation Study

| TIM | pred_len | Ours mse | Ours mae | Time_wo mse | Time_wo mae | Res_wo mse | Res_wo mae | Feat_wo mse | Feat_wo mae |
|---|---|---|---|---|---|---|---|---|---|
| ETTh1 | 96 | 0.367 | 0.391 | 0.379 | 0.398 | 0.379 | 0.397 | 0.378 | 0.396 |
| | 192 | 0.424 | 0.425 | 0.438 | 0.428 | 0.436 | 0.427 | 0.433 | 0.426 |
| | 336 | 0.472 | 0.446 | 0.493 | 0.459 | 0.481 | 0.449 | 0.482 | 0.451 |
| | 720 | 0.471 | 0.469 | 0.497 | 0.477 | 0.494 | 0.478 | 0.492 | 0.476 |
| | AVG | 0.436 | 0.435 | 0.452 | 0.440 | 0.448 | 0.438 | 0.446 | 0.437 |
| ETTh2 | 96 | 0.289 | 0.342 | 0.291 | 0.343 | 0.292 | 0.344 | 0.292 | 0.344 |
| | 192 | 0.374 | 0.393 | 0.377 | 0.394 | 0.375 | 0.394 | 0.377 | 0.394 |
| | 336 | 0.419 | 0.430 | 0.418 | 0.431 | 0.417 | 0.430 | 0.417 | 0.430 |
| | 720 | 0.427 | 0.444 | 0.431 | 0.446 | 0.432 | 0.447 | 0.430 | 0.446 |
| | AVG | 0.377 | 0.402 | 0.379 | 0.404 | 0.379 | 0.404 | 0.379 | 0.403 |
| ETTm1 | 96 | 0.315 | 0.357 | 0.320 | 0.360 | 0.318 | 0.357 | 0.327 | 0.365 |
| | 192 | 0.361 | 0.383 | 0.366 | 0.385 | 0.361 | 0.381 | 0.364 | 0.384 |
| | 336 | 0.386 | 0.402 | 0.412 | 0.411 | 0.397 | 0.405 | 0.401 | 0.408 |
| | 720 | 0.469 | 0.446 | 0.495 | 0.452 | 0.456 | 0.441 | 0.454 | 0.442 |
| | AVG | 0.382 | 0.397 | 0.398 | 0.402 | 0.383 | 0.396 | 0.387 | 0.400 |
| ETTm2 | 96 | 0.172 | 0.253 | 0.176 | 0.259 | 0.170 | 0.254 | 0.175 | 0.258 |
| | 192 | 0.233 | 0.294 | 0.234 | 0.297 | 0.238 | 0.298 | 0.238 | 0.299 |
| | 336 | 0.292 | 0.333 | 0.295 | 0.337 | 0.299 | 0.338 | 0.301 | 0.339 |
| | 720 | 0.391 | 0.392 | 0.400 | 0.398 | 0.395 | 0.395 | 0.398 | 0.396 |
| | AVG | 0.272 | 0.318 | 0.276 | 0.323 | 0.276 | 0.321 | 0.278 | 0.323 |
| electricity | 96 | 0.144 | 0.241 | 0.156 | 0.255 | 0.152 | 0.253 | 0.169 | 0.269 |
| | 192 | 0.164 | 0.259 | 0.174 | 0.271 | 0.170 | 0.269 | 0.181 | 0.275 |
| | 336 | 0.173 | 0.271 | 0.190 | 0.289 | 0.186 | 0.287 | 0.197 | 0.290 |
| | 720 | 0.205 | 0.301 | 0.219 | 0.312 | 0.213 | 0.311 | 0.234 | 0.320 |
| | AVG | 0.172 | 0.268 | 0.185 | 0.282 | 0.180 | 0.280 | 0.195 | 0.288 |
| traffic | 96 | 0.447 | 0.277 | 0.473 | 0.313 | 0.474 | 0.306 | 0.492 | 0.311 |
| | 192 | 0.458 | 0.287 | 0.474 | 0.317 | 0.482 | 0.316 | 0.506 | 0.326 |
| | 336 | 0.471 | 0.292 | 0.482 | 0.317 | 0.492 | 0.320 | 0.519 | 0.332 |
| | 720 | 0.503 | 0.310 | 0.520 | 0.344 | 0.538 | 0.342 | 0.554 | 0.343 |
| | AVG | 0.469 | 0.292 | 0.487 | 0.323 | 0.497 | 0.321 | 0.518 | 0.330 |

