# OpenReview forum: "Non-Additive Time-Series Forecasting via Cross-Decomposition and Linear Attention"
_ICLR.cc/2026/Conference — Submitted to ICLR 2026_

### Official Review · Reviewer_BPio · 2025-10-28

**Soundness:** 2
**Presentation:** 3
**Contribution:** 2
**Rating:** 4
**Confidence:** 3

**Summary:**

This paper introduces TIM, an all-MLP forecaster designed from the ANOVA/Hoeffding target. TIM consists of lightweight branches capture main effects, and a DCN-style cross stack models the orthogonal non-additive interaction subspace. TIM achieves linear time and memory complexity via axis-wise linear selfattention combined with DCN-based feature crossing, while keeping parameters comparable to compact MLP baselines.

**Strengths:**

- The paper is well-structured and easy to follow.
- The paper provides detailed theoretical analysis, including degree and rank guarantees for the cross stack, as well as risk decomposition identities that explain the additive error gap. These contributions ground the proposed method in a strong mathematical framework.
- The proposed TIM model balances accuracy and computational efficiency, outperforming Transformer-based models while maintaining a lightweight, all-MLP architecture. This makes it particularly suitable for resource-constrained environments.

**Weaknesses:**

- Although TIM achieves SOTA results on several datasets, its performance on certain benchmarks (e.g., Traffic, Solar-Energy) shows only marginal improvements.
- While the paper compares TIM against several Transformer-based and MLP-based methods, there are some latest and relevant works omitted. For example, Transformer-based TimeXer [1] and MLP-based TimeMixer++ [2] and SOFTS [3].

[1] Wang, Yuxuan, et al. "Timexer: Empowering transformers for time series forecasting with exogenous variables." Advances in Neural Information Processing Systems 37 (2024): 469-498.

[2] Wang, Shiyu, et al. "Timemixer++: A general time series pattern machine for universal predictive analysis." arXiv preprint arXiv:2410.16032 (2024).

[3] Han, Lu, et al. "Softs: Efficient multivariate time series forecasting with series-core fusion." Advances in Neural Information Processing Systems 37 (2024): 64145-64175.

- While the paper highlights TIM’s interpretability through its component-wise attributions and decomposition regularizer, the explanation and evaluation of this interpretability are limited.

**Questions:**

Please refer to weakness

---

### Official Review · Reviewer_3m5n · 2025-10-29

**Soundness:** 2
**Presentation:** 2
**Contribution:** 2
**Rating:** 2
**Confidence:** 3

**Summary:**

This paper introduces TIM (Time-series Interaction Machine), a fully-MLP time series forecasting model based on ANOVA/Hoeffding decomposition. It enhances long-term multivariate forecasting performance by explicitly modeling main effects and non-additive interaction effects. The core innovation is explicit separation of forecasting signals into three components: (1) main temporal effects via Time Fusion, (2) non-additive multivariate interactions via Feat Fusion using DCN-style cross stacks, and (3) residual corrections via Res Fusion. Experiments on 8 benchmarks show TIM achieves 25/48 first places in MSE and 21/48 in MAE, outperforming DLinear, PatchTST, TimeMixer, and iTransformer.

**Strengths:**

- This paper addresses a genuine limitation in existing time series forecasting: most models implicitly emphasize additive effects while inadequately modeling interaction effects (cross-variable and cross-temporal dependencies).
- The study is grounded in a rigorous mathematical framework (e.g., ANOVA/Hoeffding decomposition). It provides comprehensive theoretical guarantees through a series of theorems and corollaries, including error bounds, interaction purity, and coefficient formulations.
- The experiments cover 8 benchmark datasets across diverse domains and evaluate 4 prediction horizons (short-term and long-term). The model competes against 7 SOTA methods and achieves strong performance, with 25/48 first places in MSE and 21/48 in MAE metrics.

**Weaknesses:**

- The theoretical framework places significant emphasis on exogenous variables. However, the experimental section does not clarify whether the datasets used incorporate these variables or how they are accounted for. Moreover, no evaluation is provided regarding the actual impact of these variables on the results.

- While the paper claims to contribute to model interpretability on component-wise attribution and cross-term explanations, it offers no empirical evidence or illustrative examples in the experiments to substantiate these claims.

- A more thorough and in-depth analysis of the experimental results should be conducted. This could include, for example, error analysis, results on different settings, or an examination of how well the results align with theoretical expectations.

**Questions:**

Please refer to weakness.

---

### Official Review · Reviewer_qG8Q · 2025-10-31

**Soundness:** 3
**Presentation:** 3
**Contribution:** 3
**Rating:** 6
**Confidence:** 4

**Summary:**

This paper proposes the Time Series Interaction Machine (TIM), an MLP-based forecaster for non-additive time series forecasting that explicitly aligns with ANOVA/Hoeffding decomposition. The key innovation is the use of cross decomposition and linear attention to model non-additive interactions in time series data. The architecture consists of three main components: (1) a lightweight branch capturing main effects, (2) a Feat Fusion module using Deep Cross Network (DCN)-style cross stacks to extract multivariate interaction effects, and (3) a linear transformer backbone with channel-independent structure. The paper provides theoretical justification by connecting the architecture to ANOVA decomposition, arguing that the learned components correspond to theoretical main effects and interaction effects. Experiments are conducted on standard datasets against SOTA baselines.

**Strengths:**

## Strengths

1. **Well-written introduction and background**: The first three paragraphs of the introduction provide clear and intuitive background. The abstract paints a good high-level picture of the approach with main effects and interaction effects.

2. **Novel contribution in Feat Fusion**: The Feat Fusion module is the most impressive innovation. It effectively captures multivariate interaction effects, which distinguishes this work from previous methods that can be viewed as "main effect only" models.

3. **Clear presentation aids**: Figure 1 is very helpful in understanding the architecture. Algorithm 2 is useful and clear, aligning well with Figure 1. Lines 191-211 describing the fusion mechanisms are intuitive and easy to read.

4. **Theoretical grounding**: The connection to ANOVA decomposition provides theoretical justification for the architecture design, with learned components corresponding to theoretical components.

5. **Solid experimental setup**: Standard datasets and SOTA baselines are used appropriately.

6. **Complementary approach**: The work complements recent developments in transformer-based time series forecasting methods and appears modular—the transformer backbone could potentially be switched.

**Weaknesses:**

## Weaknesses

1. **Clarity issues in the abstract**:
   - Acronyms (DCM, CP rank) are not spelled out initially
   - Confusion about how "MLP forecaster" and "linear self-attention" fit together
   - Lines 19-20 become difficult to follow

2. **Organizational and readability issues**:
   - References in Section 2.1 (page 2) point to pages 5-6, disrupting reading flow
   - Section 3 feels abrupt; unclear transition from method description to theoretical alignment
   - Purpose of Sections 3.4, 3.5, 3.6, and 3.7 is unclear—are these all theoretical remarks?

3. **Insufficient justification for architectural choices**:
   - The role and necessity of residue branches (Res Fusion) is not well explained. Why is it essential beyond being "like a skip connection"?
   - Line 190: unclear how Time Fusion, Feat Fusion, and Rest Fusion "share the same architecture" when designed for different purposes
   - Line 240: confusing whether regression targets relate to loss function construction or component interpretation

4. **Weak ablation study**: The "without RES" ablation confirms RES helps performance, but:
   - No clear high-level intuition for why RES is absolutely needed
   - Line 397 mentions RES captures cross-variable dependencies, but doesn't clarify its exact unique contribution
   - Line 403 mentions performance without RES but doesn't justify its necessity

5. **Missing key experiments**: A more comprehensive ablation study is needed to demonstrate the value of Feat Fusion across different backbones:
   - TIM + DLinear vs. DLinear alone
   - TIM + PatchTST vs. PatchTST alone
   - TIM + iTransformer vs. iTransformer alone

   This would better isolate the contribution of interaction effect modeling from the choice of backbone.

**Questions:**

I have listed structured questions (with help of LLM) in the above weakness part. I am going to say here my honest thoughts when reading the paper as it presents, and hopefully this can help you understand how a new reader perceives your paper. These raw feelings are genuine and I hope they provide a more human-to-human communication and contexts for the structured question above.

# Review: Non-Additive Time Series Forecasting

## Abstract

The review for non-additive time series forecasting via cross decomposition and linear attention. The abstract actually makes sense. It is trying to model the non-additive interactions and use a design from ANOVA decomposition. It paints an intuitive high-level picture of what it is, with main effects and interaction effects.

Minor points: the acronyms, although standard, are used a lot. You can spell them out.

I'm also confused: it mentions "MLP forecaster" earlier in the abstract, and then in the middle part of the abstract, it says "access via linear self-attention." So I don't see how the linear self-attention plugs in here. That's okay, I can wait.

I look forward to the transparent cross-term explanations.

In the middle of the abstract (the line starting around line 19-20), I start to get a little bit lost, but that's okay—I can read the main paper.

## Introduction

The first two paragraphs of the background in the introduction are really good. The third paragraph is also very good. Basically, it gives the background pretty well.

The proposed method is called Time Series Interaction Machine. It's an MLP forecaster explicitly aligned with ANOVA/Hoeffding decomposition. Interesting. A lightweight branch captures the main effect, and a DCN-style cross stack.

DCN means Deep Cross Network. The mathematics is pretty simple—it's basically an MLP path that learns implicit nonlinear interaction effects.

The criticism of transformers is correct regarding their quadratic time and memory complexity. There are linear transformers available. But the authors are using linear transformers. Otherwise, I have no complaint about the introduction. It's pretty good.

A lot of the references in page 2 of Section 2.1 go to pages 5 and 6, making it a little bit difficult to read. But it's just a minor point. For better smoothness, I'm actually going to glance through 2.1 without paying too much attention, and move on to 2.2.

## Method

It's already using linear attention in line 94. So that's good. The interpretation of a kernel smoother is established in the literature (line 100).

So the key in Section 2.1.4 should be multivariate interaction features that are learned. I didn't see in the introduction right here how linear attention is used. Or is linear attention just used as a benchmark?

Alright, I see. Figure 1 makes this really helpful. I'm assuming in Figure 1 the Feature Feat Fusion is the main innovation. Now, the extra multivariate interaction effects—yes, like cross-vector, like DCN in time. And component (b) is a linear transformer backbone going through a channel-independent structure. I see.

So really, the new thing here is the first block—the Feat Fusion that extracts multivariate interaction effects. Yes, so in that sense, the time fusion and transformer backbone previously can be thought of as a main effect only model. And this work is adding the Feat Fusion to add interaction effects. Very interesting.

I don't quite see what the residue branches are doing. It seems that it's just capturing anything not in the main model as residue, kind of like a skip connection or something like that. I would be curious about how important the Res Fusion is, and why it is essential.

Interesting. So because Time Fusion and Res Fusion are per-variable channel-independent, the multivariate effect is solely captured in the Feat Fusion.

I'm a little confused. In line 190, how can Time Fusion, Feat Fusion, and Rest Fusion share the same architecture? They are designed for very different things. In what sense are they sharing the architecture?

From lines 191 to 211, the goal and the interpretation of those fusion mechanisms are intuitive and easy to read. Algorithm 2 is useful and clear for the overall flow. It aligns well with Figure 1.

## Theoretical Connections

I feel Section 3 is a little bit sudden. I just thought that I had finished reading the method, and then there seems to be some alignment that needs to be done. I'm a little bit confused.

Also, in line 240, it starts to introduce regression targets, which confuses me. Is it about loss function construction, or is it about the interpretation of each component?

I get Section 3.2—the decomposition is somehow theoretically justified. Is the claim that the architecture mirrors this ANOVA decomposition, and therefore, the learned components will have a good correspondence to the theoretical components? Is that the story?

Oh, that seems to be the case. After reading line 278, it seems that the cross-branch and main effect correspond to the earlier architecture.

So I don't quite get what's the purpose of presenting Section 3.4, Section 3.5, and Section 3.6, together with Section 3.7. They are all remarks and connections to some theoretical study—theoretical interpretation. Is that right?

## Experiments

Now I move on to Section 4, the experiment session. I can see that the dataset descriptions are all standard datasets. The SOTA baselines are also standard.

The ablation study actually includes "without RES" That confirms my intuition that the residue seems to be least coherent to the story. I get that it helps with performance. Is there any discussion on the high-level intuition about why the residue is absolutely needed?

Like you mentioned in line 397, “RES processes univariate time series as tokens and captures cross-variable dependencies." Therefore, what is the exact contribution of RES? That's a good place for you to remind me.

Yes, in line 403, you actually mentioned why the "without RES" setting seems to be still ok. But you didn't remind me why the RES connection is absolutely needed.

## Overall Assessment

Overall, I think it's an interesting paper. From a different angle, I can see that it can complement a lot of the recent developments in time series forecasting methods, especially transformer-based time series forecasting methods. This Feat Fusion does seem to capture interaction effects.

It also appears to me that you can actually switch the transformer backbone. So for the ablation study, I would say a more fair ablation study would be: if you're using DLinear as the backbone, then you have TIM plus DLinear; TIM applied on DLinear, see how good it is. Use PatchTST as the backbone, and then TIM plus PatchTST, see how good it is. Or iTransformer as the backbone. You get what I'm saying. I think that would illustrate how important the interaction effect and this residue prediction really are. Because I really think the Feat Fusion is the most impressive new thing here.

Overall, I think this is a good paper. An interesting one. I wouldn't say the entire thing is—the entire thing is a solid read. There's some confusion, but there's one part of the idea that I really like. I'm not too convinced about the Time Fusion and the Res Fusion, but I really like the Feat Fusion perspective.

---

### Official Review · Reviewer_ScVg · 2025-11-01

**Soundness:** 1
**Presentation:** 1
**Contribution:** 1
**Rating:** 2
**Confidence:** 4

**Summary:**

This paper proposes TIM, which employs the decomposition of features, time, and residuals, to effectively capture both additive and interaction effects with minimal computational overhead.
The paper's experiments demonstrate that TIME outperforms classical time series forecasting baselines in terms of both accuracy and efficiency across multiple datasets.

**Strengths:**

This paper proposes TIM, which employs the decomposition of features, time, and residuals, to effectively capture both additive and interaction effects with minimal computational overhead.
The paper's experiments demonstrate that TIME outperforms classical time series forecasting baselines in terms of both accuracy and efficiency across multiple datasets.

**Weaknesses:**

1. The description of the model architecture in the paper is disorganized, and it is unclear how each module actually works. There is a lack of complete mathematical formulation, and the architectural description is not clear enough.

2. The paper claims to use Feat Fusion for capturing the non-additive (interaction) component, Time Fusion for capturing temporal shifts and main additive effects, and Res Fusion for capturing residual structure not explained by main or interaction effects. However, there are no corresponding designs to ensure these defined functions.

3. The paper has poor readability, with many terms not clearly defined. For example, DCN-style and CP rank.

4. The paper's innovation is limited. AXIS-WISE LINEAR SELF-ATTENTION seems to be directly adapted from Linear attention, yet there are no relevant citations.

5. The paper's experiments are not convincing with outdated baselines.

6. The paper's layout is disorganized, with chaotic table formatting. The ablation experiment table 4 is in the appendix, but there is no proper reference to the appendix.

7. The method description in the paper is very unclear, and the source code is not provided, making both the results and the methods irreproducible.

**Questions:**

Please refer to the weakness.

---

### Meta-Review · Area_Chair_yL8u · 2026-01-06

**Summary:**

The reviewers have pointted out four key waeknesses preventing this paper to be accepted.

(1) Writting and organization: the description of the model architecture in the paper is disorganized, and it is unclear how each module actually works. There is a lack of complete mathematical formulation, and the architectural description is not clear enough;
(2) Experiments: The experimental results are deemed less persuasive due to the use of outdated baselines and the absence of comparisons with recent state-of-the-art (SOTA) models;
(3) Technical details: While TIM emphasizes interpretability via component-wise attribution, the current qualitative and quantitative evaluations of these claims are considered limited;
(4) Limited novelty: the innovation of the work is limited, specifically regarding the "Axis-wise Linear Self-Attention," which appears to be directly adapted from Linear attention, yet there are no relevant citations.

Given the above major, I recommend rejecting this paper.

**Reviewer Concerns:**

The authors did not provide rebuttal.

**Reviewer Scores:**

The authors did not provide rebuttal.

---

### Decision · Program_Chairs · 2026-01-26

Reject